# The persistent benefits of decreasing default pill counts for postoperative narcotic prescriptions

Nathan Coppersmith[1]☯*, Joshua Sznol[1]☯, Andrew Esposito[1], Emily Flom[1], Alexander Chiu[2], Peter Yoo[3]

1 Department of Surgery, Yale School of Medicine, New Haven, Connecticut, United States of America,
2 Department of Surgery, University of Wisconsin-Madison, Madison, Wisconsin, United States of America,
3 Academic Affairs, Hartford Healthcare, Hartford, Connecticut, United States of America

☯ These authors contributed equally to this work.
* nathan.coppersmith@yale.edu

**Data Availability Statement:** All relevant data are within the manuscript and its Supporting information files.

## Abstract

### Background

In 2017, a university-based academic healthcare system changed the opioid default pill count from 30 to 12 pills. Modifying the electronic default pill count influences short-term clinician prescribing practices. We sought to understand the long-term impact on postoperative opioid prescribing habits after an opioid default pill count reduction.

### Materials and methods

A retrospective electronic medical record system (EMRS) review was conducted in a healthcare system comprised of seven affiliated hospitals. Patients who underwent a surgical procedure and were prescribed an opioid on discharge between 2017–2021 were evaluated. All prescriptions were converted into morphine equivalents (MME). Analyses were performed with the chi-square test and Bonferonni adjusted t-test.

### Results

191,379 surgical procedures were studied. The average quantity of opioids prescribed decreased from 32 oxycodone 5 mg tablets in 2017 to 21 oxycodone 5 mg tablets in 2021 (236 MME to 154 MME, p<0.001). The percentage of patients obtaining a refill within 90 days of surgery varied between 18.3% and 19.9% (p<0.001). Patients with a pre-existing opioid prescription and opioid-naïve patients both had significant reductions in prescription quantities above the default MME (79.7% to 60.6% vs. 65.3% to 36.9%, p<0.001). There was no significant change in refills for both groups (pre-existing 36.7% to 38.3% (p = 0.1) vs naïve 15.0% to 15.3% (p = 0.29)).

### Conclusions

The benefits of decreasing the default opioid pill count continue to accumulate long after the original change. Physician uptake of small changes to default EMRS practices represents a

**Funding:** The author(s) received no specific funding for this work.

**Competing interests:** The authors have declared that no competing interests exist.

sustainable and effective intervention to reduce the quantities of postoperative opioids prescribed without deleterious effects on outpatient opiate requirements.

## Introduction

Prior research has demonstrated variable and excessive opioid prescribing habits after general surgery procedures [1]. Further work has revealed that the risk of patients developing new persistent opioid use disorder after surgery is increased with the size of opioid prescription [2]. In an effort to reduce the volume of unnecessary and problematic opioid prescriptions, multiple studies have demonstrated that decreasing the auto-populated default number of pills to be prescribed in the electronic medical record system (EMRS) leads to decreased opioid prescription quantities [3, 4].

Decreasing quantities of opioids prescribed after surgery raised concerns about adequacy of analgesia and the clerical burden of refilling inadequate prescriptions. However, it has been established that despite reductions in the size of opioid prescriptions after a default change, there have not been significant increases in the prescription refill rates and no changes in the proportion of patients who reported using half or less of the total opioids prescribed [3, 5]. Additional work evaluating outcomes has not found a relationship between patient satisfaction after surgery and opioid prescription size, and there were similar levels of high satisfaction when patients who were prescribed opioids after surgery were compared to those who were not prescribed any opioids [6, 7]. Together, this evidence supports the minimization of postoperative opioid prescriptions without causing deleterious effects on patient postoperative pain control.

Multiple national and state level measures have been introduced to address the opioid use epidemic [8, 9]. Efforts to reduce the opioid epidemic from a policy level have had mixed results, but guideline and policy implementations were noted to reduce opioid prescription use by 15% and reduce multiple-provider episodes by up to 62%. Overall, the daily dose of opioid prescriptions decreased by 16.9% from 2010 to 2015 [10]. However, the total amount of opioids prescribed in 2015 was still three times higher than in 1999 [10] and recent clinical management pathways have sought to reduce or eliminate opioid medications from postoperative care when appropriate [11].

The three-month impact of an EMRS default change from thirty pills to twelve pills in a large academic health system comprising both independent and academic centers was previously studied by Chiu et al. and showed a significant reduction in postoperative opioid prescriptions after the default change without a concurrent increase in refills [3]. That study examined patients undergoing the ten most common operations across the healthcare system who were discharged by postoperative day 1. However, the long-term impact of this same, or any, default pill count change on prescribing practices over multiple years has not been examined. Therefore, it is not clear if a default pill count change represents an intervention that produces long-term effects towards reducing excessive opioid prescribing practices, or if prescribing practices regress over time back to habits that existed prior to the intervention. In this study, we evaluated the long-term effects of modifying opioid defaults in a large academic health system comprising both independent and academic centers.

## Materials and methods

The default pill count was changed in the EMRS from 30 to 12 pills in May 2017. 12 pills was chosen at the time in order to provide a 5 mg oxycodone immediate relief tablet every six

hours for three days. All patients who underwent a surgical procedure and were prescribed an opioid on discharge regardless of discharge date between May 2017 and December 2021 were evaluated in a retrospective review at a large academic health system comprising seven independent and academic hospitals. Both inpatient and outpatient surgeries were included. Opioid prescriptions included all formulations that contained codeine, hydrocodone, hydromorphone HCl, morphine sulfate, oxycodone, and tramadol HCl. Demographics, surgical procedure, admissions characteristics, active outpatient narcotic prescriptions, and refills within 90 days of discharge were obtained. A 90-day refill cut-off was chosen to coincide with the Federal Controlled Substances Act [12]. The data were accessed for research purposes on May 9, 2022 and July 17, 2022. The raw data were pulled from the electronic medical record system by the university data analytics team in accordance with university policy. This study was deemed exempt by the Yale University Institutional Review Board. The Yale University Institutional Review Board waived the requirement for informed consent for this study. The authors had access to information that could identify participants during data collection and the data was anonymized for analysis.

All prescriptions were converted into morphine equivalents (MME) using the United States Centers for Disease Control and Prevention morphine milligram equivalent doses [13]. Prescriber narcotic patterns were categorized as above, equal to, or below the default MME (90 MME). The default of 90 MME is equivalent to 12 oxycodone 5 mg immediate-release tablets. Patients were categorized as opiate naïve if they did not have an active outpatient opiate prescription on admission. Univariate statistics were calculated for the sample. Comparisons between opioid naive status, prescription year, and prescriber pattern were compared with chi-square analysis. Comparisons between each year were performed with a Bonferonni adjusted t-test. SAS Enterprise version 9.4 was used for statistical analyses.

## Results

### Demographics

Between May 2017 to December 2021, 191,379 surgical procedures were studied. The patient demographics are provided in Table 1. Patients were discharged home or to hospice after 91.7% of the procedures. 17.95% of patients had a pre-existing opioid prescription.

### Impact of default change

From 2017–2021, the average MME of opioids prescribed on discharge decreased from 236 to 154 (p<0.001) (Fig 1). In terms of 5 mg oxycodone immediate-release tablets, this translates to a decrease from an average prescriptions size of 32 oxycodone tablets in 2017 to an average prescription size of 21 oxycodone tablets in 2021. Additionally, the proportion of opioids prescribed greater than the default MME decreased from 67.9% in 2017 to 41.6% in 2021, the proportion prescribed at the default increased from 14.2% to 23.8%, and the proportion prescribed below the default increased from 17.8% to 34.6% (p<0.001) (Fig 2).

The percentage of patients over the five years obtaining a refill within 90 days of surgery varied between 18.3% and 19.9% (Table 2), (p<0.001). The average number of patients with prior opioid prescriptions also varied by year, ranging from 16.3% to 19.8% (p<0.001).

### Opioid naïve versus pre-existing opioid prescription patients

A subgroup analysis of patients with a pre-existing opioid prescription compared to opioid-naïve patients showed that both groups had significant reductions over this time period in prescription quantities above the default MME (79.7% to 60.6% vs. 65.3% to 36.9%, p<0.001)

**Table 1. Patient demographics.**

| | Percent of Patients (*N* = 191,379) |
|---|---|
| **Age** | |
| Average | 51.1 years (±19.6) |
| **Gender** | |
| Female | 59.8% (114,411) |
| Male | 40.2% (79,968) |
| **Length of Stay** | |
| Average | 2.7 days (±6.02) |
| **Discharge Disposition** | |
| Home/Hospice | 91.7% (175,206) |
| Against Medical Advice | 0.04% (81) |
| Other Healthcare | 8.3% (15,786) |
| **Race** | |
| American Indian or Alaska Native | 0.3% (583) |
| Asian | 2.1% (3,913) |
| Black or African American | 14.1% (27,015) |
| Native Hawaiian or Other Pacific Islander | 0.2% (464) |
| Other | 12.9% (24,691) |
| White or Caucasian | 70.4% (134,672) |
| **Ethnicity** | |
| Hispanic or Latino | 15.6% (29,498) |
| Non-Hispanic | 84.4% (159,652) |
| **Prior Opioid Use** | |
| Opioid Naïve | 82.1% (157,027) |
| Pre-existing Opioid Prescription | 17.95% (34,352) |

(Table 3). This decrease in prescriptions greater than the default quantity was matched with significant increases in prescriptions at or below the default quantity (Table 3). Despite the reduction in overall prescription size, patients with a pre-existing opioid prescription were consistently prescribed higher quantities of opioids compared to opioid-naïve patients.

Patients with a pre-existing opioid prescription also had a consistently higher 90-day prescription refill rate. The percentage of patients with a pre-existing opioid prescription who obtained refills changed from 36.7% to 38.3% from 2017 to 2021, although this trend did not reach significance (p = 0.1). For opioid naïve patients, the percentage of refills slightly increased from 15.0% to 15.3%, although this increase was not statistically significant (p = 0.29). For both groups of patients, the overall prescription quantities decreased, but the rates of prescription refills were stable during the study period.

## Discussion

This study demonstrates the long-term effect of decreasing the default pill count in an electronic medication record system (EMRS). Prior research has only evaluated the short-term impact of default changes and this study provides the novel finding that a default pill count change leads to sustained and profound effects on prescriber practices across a healthcare system over multiple years. The five years since the default change saw a 35% reduction in the average postoperative opioid prescription size, which represents a large quantity of opioid pills that were not prescribed. While the risk of developing a new persistent opioid use after surgery is small, for opioid-naïve patients, larger post-operative prescriptions are associated with an

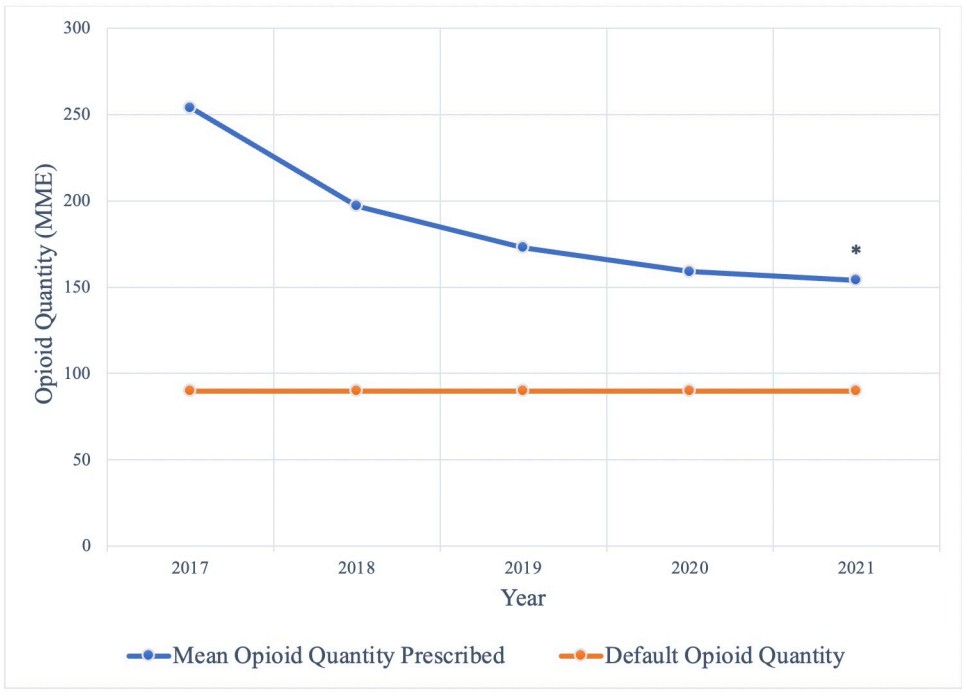

**Fig 1. Mean postoperative opioid quanities prescribed.** The mean postoperative opioid quantity prescribed (MME) from 2017–2021 compared to the default opioid quantity (MME), * = *p*<0.001.

increased risk of developing a new persistent opioid use [14] and reducing the default pill count and reducing post-op opioid prescription sizes can hopefully help to mitigate this risk.

The trends we saw in our data mirror those seen in Connecticut during the same time period. Data from the Connecticut Prescription Monitoring and Reporting System (CPMRS)

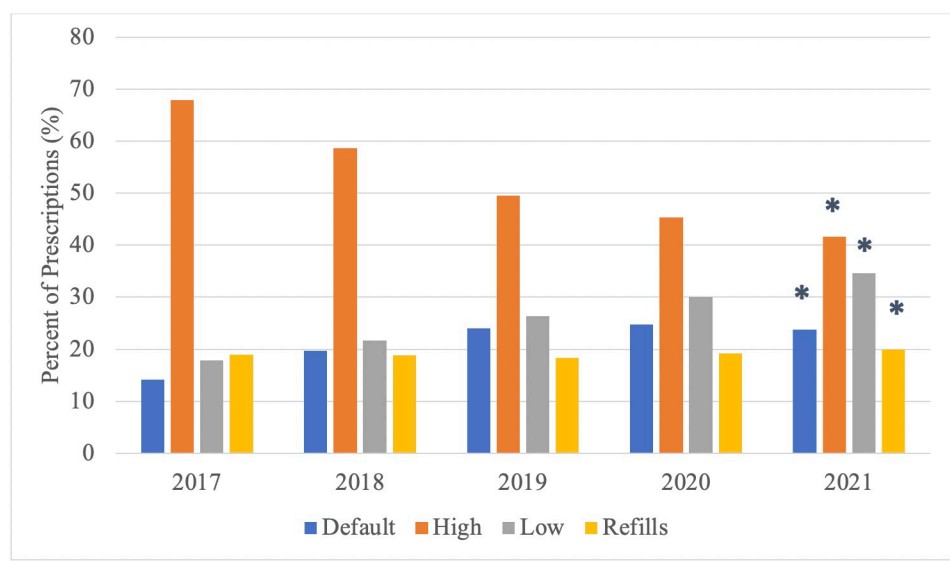

**Fig 2. Comparing quantities of postoperative opioids prescribed.** The percent of postoperative opioid prescriptions at the default MME, higher than the default MME, lower than the default MME, and resulting in refills from 2017–2021, * = *p*<0.001.

**Table 2. The raw quantities of 90-day opioid refills and the 90-day refills as a percentage of total annual prescriptions.**

| Year | Raw Quantity of 90-Day Refills | 90-Day Refills as a Percentage of Total Prescriptions |
|---|---|---|
| 2017 | 6,223 | 19.0% |
| 2018 | 8,285 | 18.8% |
| 2019 | 7,672 | 18.3% |
| 2020 | 6,564 | 19.2% |
| 2021 | 7,616 | 19.9% |

system shows that in Connecticut, the number of opioid prescriptions in the state fell from 2,169,959 in 2017 to 1,731,539 in 2021 [15]. This represents a 20% reduction in the number of opioid prescriptions in 2021 compared to 2017. The quantity of opioid prescriptions in Connecticut for greater than 90 MME also fell from 8% in 2017 [16] to 2% in 2021 [17], suggesting that clinicians are prescribing fewer opioids. Interestingly, the largest decrease in prescription quantity was between 2017 and 2018 and then the magnitude of the decrease became smaller each year afterwards, suggesting that the average prescription quantity could be approaching almost an asymptotic relationship with the default quantity. This asymptote may represent the appropriate mix of prescription quantities for different patients to receive, and may in part reflect the provider comfort with specific quantities of pills relative to the default.

Despite the steadily decreasing quantities of opioid pills prescribed, the refill rate remained stable over the five years, suggesting that patients were not under-prescribed pain control medications even as the quantities of pills prescribed continued to decrease. That said, the data did demonstrate a significant rebound in the quantities of 90-day prescription refills. After an initial decrease in the number of opioid prescription refills from 2019 to 2020, there was an increase in refills in 2021. We hypothesize that this fluctuation in raw quantities of refills was due to the decrease in surgeries in 2020 at the start of the COVID-19 pandemic, with a

**Table 3. Opioid prescription size relative to the default quantity between patients who had a pre-existing opioid prescription (prior opioid) and patients who were opioid naïve and the 90-day prescription refill rate of prior opioid patients and opioid naïve patients.**

| Prescription Size Relative to Default Quantity | 2017 Percent of Prescriptions (N) | 2021 Percent of Prescriptions (N) | Change in Percentage | p-value |
|---|---|---|---|---|
| Greater than Default *Prior Opioid* | 79.7% (4,847) | 60.6% (4,604) | -19.1% | p<0.001 |
| Greater than Default *Opioid Naïve* | 65.3% (17,383) | 36.9% (11,331) | -28.4% | p<0.001 |
| At Default *Prior Opioid* | 10.6% (643) | 17.5% (1,332) | +6.9% | p<0.001 |
| At Default *Opioid Naïve* | 15.0% (4,007) | 25.3% (7,759) | +10.3% | p<0.001 |
| Less than Default *Prior Opioid* | 9.7% (592) | 21.9% (1,662) | +12.2% | p<0.001 |
| Less than Default *Opioid Naïve* | 19.7% (5,246) | 37.8% (11,595) | +18.1% | p<0.001 |
| | 2017 90-Day Prescription Refill Rate | 2021 90-Day Prescription Refill Rate | Change in Percentage | p-value |
| Refill Rate *Prior Opioid* | 36.7% | 38.3% | +1.6% | p = 0.1 |
| Refill Rate *Opioid Naïve* | 15.0% | 15.3% | +0.3% | p = 0.29 |

subsequent increase in surgeries again in 2021. Despite the changes in raw quantities of refills each year from 2017–2021, refills as a percentage of total prescriptions each year only fluctuated by 1.6% from each other at most. We believe that this demonstrates that there was not great change in the actual proportion of patients who received a refill each year.

The sub-analysis also showed that patients with a pre-existing opioid prescription were consistently more likely to be prescribed quantities of opioids above the default and to have a higher refill rate, suggesting that their opioid requirements are understandably different than opioid-naïve patients. Patients with prior opioid use have been found to have a higher risk of developing persistent postoperative opioid use and can have worse postsurgical outcomes compared to opioid naïve patients, suggesting that these patients require particular attention in the perioperative period [18, 19]. However, patients with pre-existing opioid prescriptions had decreasing quantities of opioid prescriptions across the span of five years, similar to the opioid-naïve cohort. Both of these reductions in prescription sizes were not accompanied by significant changes in the 90-day prescription refill rate.

This study has several limitations. While our conclusions utilize administrative data and inferences based on the data, this study lacks patient-reported outcomes regarding pain control and satisfaction. We also did not assess for the use of non-opioid pain control and how those prescribing practices could have impacted opioid prescribing habits. The past several years have also seen increased awareness of the opioid epidemic and the dangers of opioid overprescription. We were not able to quantify the extent to which other potential interventions to reduce opioid overprescription and increased awareness of overprescription may have impacted prescribing habits. That said, no intervention towards reducing opioid overprescription across this healthcare system was as broad reaching across sites and providers as the EMRS default pill count change.

These results show that physician uptake of small changes to default EMRS practices continues to grow over time and default changes to opioid prescription quantities represent a sustainable and effective intervention to reduce the quantities of postoperative opioids prescribed without deleterious effects on outpatient opiate requirements. The default change is a relatively blunt instrument to impact prescribing habits, but it is effective in pushing broad swaths of prescribers towards lowering quantities of opioid prescriptions. The use of multimodal analgesia at different points in the perioperative period has been shown in multiple clinical scenarios to reduce the use of postoperative opioids [20–23] and incorporating multimodal analgesia plans into surgical patient care can pair with a lower default count to reduce the quantities of postoperative opioids needed. In the future, default changes could be combined with prescribing recommendations and multimodal analgesia plans targeted to specific patients and procedures in order to find the optimal pain control for each individual patient and operation.

## Supporting information

**S1 Table. Patient demographics.** Patient demographics as overall and broken down by year.
(DOCX)

**S2 Table. Quantities of postoperative opioids prescribed.** The raw count and percent of postoperative opioid prescriptions at the default MME, higher than the default MME, lower than the default MME, and resulting in refills from 2017–2021 based on prior opioid use or opioid naïve.
(DOCX)

**S3 Table. Quantities of postoperative opioid refills prescribed.** The raw quantities of 90-day opioid refills and the 90-day refills as a percentage of total annual prescriptions based on prior

opioid prescription or opioid-naïve patients.
(DOCX)

**S1 File. Postoperative opioid prescription raw data file.** The raw data file that contains year of surgery, demographic information, length of stay and opioid prescription data.
(XLSX)

## Author Contributions

**Conceptualization:** Nathan Coppersmith, Andrew Esposito, Alexander Chiu, Peter Yoo.

**Data curation:** Nathan Coppersmith, Joshua Sznol, Emily Flom.

**Formal analysis:** Nathan Coppersmith, Joshua Sznol, Andrew Esposito.

**Investigation:** Nathan Coppersmith, Andrew Esposito, Emily Flom, Alexander Chiu, Peter Yoo.

**Methodology:** Nathan Coppersmith, Joshua Sznol, Andrew Esposito, Emily Flom, Alexander Chiu, Peter Yoo.

**Project administration:** Nathan Coppersmith.

**Resources:** Nathan Coppersmith, Peter Yoo.

**Supervision:** Peter Yoo.

**Visualization:** Nathan Coppersmith, Joshua Sznol.

**Writing – original draft:** Nathan Coppersmith.

**Writing – review & editing:** Joshua Sznol, Andrew Esposito, Emily Flom, Alexander Chiu, Peter Yoo.

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
