## [Decision Letter · Decision Letter 0]

5 Mar 2024

PONE-D-23-43929The Persistent Benefits of Decreasing Default Pill Counts for Postoperative Narcotic PrescriptionsPLOS ONE

Dear Dr. Coppersmith,

Thank you for submitting your manuscript to PLOS ONE. After careful consideration, we feel that it has merit but does not fully meet PLOS ONE’s publication criteria as it currently stands. Therefore, we invite you to submit a revised version of the manuscript that addresses the points raised during the review process.

This is an interesting piece of work demonstrating how simple attitudes, such as reducing the number of prescribed opioid tablets in the postoperative period, can result in lower long-term usage risks. I find the study to be well-designed and presented; however, there is a need to enhance the discussion by comparing the obtained results with factors determining postoperative chronic opioid usage, as well as local characteristics of opioid abuse found in official local statistics and any available literature. I believe the discussion is incomplete and does not fully reflect the interesting findings demonstrated.

I also request that you pay attention to the reviewers' comments and respond to all queries if you agree to resubmit the study for review.

We look forward to receiving your revised manuscript.

Kind regards,

Guilherme Antonio Moreira de Barros, M.D., M.Sc., Ph.D

Academic Editor

PLOS ONE

Reviewers' comments:

Reviewer's Responses to Questions

**Comments to the Author**

1. Is the manuscript technically sound, and do the data support the conclusions?

Reviewer #1: Partly

Reviewer #2: Yes

2. Has the statistical analysis been performed appropriately and rigorously? 

Reviewer #1: Yes

Reviewer #2: N/A

3. Have the authors made all data underlying the findings in their manuscript fully available?

Reviewer #1: Yes

Reviewer #2: Yes

4. Is the manuscript presented in an intelligible fashion and written in standard English?

Reviewer #1: Yes

Reviewer #2: Yes

5. Review Comments to the Author

Reviewer #1: Dear author, although this work is very interesting and important to the scientific community, specially dealing with opioid addiction, there are some considerations that I would like to highlight:

Opioid consumption must be related to the type of surgery. I have not seen, in the table, this variable. Probably, would be a heterogeneous group, and tissue trauma can vary as well as the intensity of pain . In addition, there was no control of the use of other analgesics that could increase or reduce opioid consumption. Besides that, gender, race, social, educational and economic level, anxiety and other psychological and psychiatric comorbidities and other variables were not controlled.

There was no mention of risk stratification and the impact on opioid consumption

I think that the use of opioid have to be a rational, based on specific circumstances, and you have to discuss and suggest how to deal with a well done postoperative analgesia . Reducing the number of pills, or the the need for more over time can not be seen as an isolated event nor as a solution to the opioid crisis

Reviewer #2: The manuscript describes a practical synthesis of effect simply reducing the number of opioid analgesic pills in the treatment of postoperative pain in the period 2017 until 2021.

Some questions about this paper

Any rational for reducing 30 pills to 12 pills? why not reduces 30-50% to opioids pill? Can you explain us more?

Page 5: Did not mention the calculation for converting each analgesic to morphine (MME), as depending on the reference, the values vary for the calculations.

Page 6: Some studies/papers report a cut off of 80-90 MME, why did they choose 90 MME? There is no consensus among the opioids guidelines as to which dose of morphine would be a cut off, most cite 90 mme

Page 6: It was not mentioned which surgeries were included in the research, we know that there are surgeries that have greater painful potential than others

As a suggestion too, include the percentages of outpatient surgeries or separate them by surgical specialty

6. PLOS authors have the option to publish the peer review history of their article (what does this mean?). If published, this will include your full peer review and any attached files.

Reviewer #1: No

Reviewer #2: No

---

## [Author Response · Author response to Decision Letter 0]

24 Apr 2024

Yale School of Medicine

333 Cedar Street 

New Haven, CT 06510

April 1, 2024

Guilherme Antonio Moreira de Barros, M.D., M.Sc., Ph.D

Academic Editor

PLOS ONE

Dear Dr. Moreira de Barros, 

Thank you for the opportunity to submit revisions to this manuscript, entitled “The Persistent Benefits of Decreasing Default Pill Counts for Postoperative Narcotics Prescriptions.” 

The review provided very helpful thoughts and suggestions, we are grateful for the excellent comments and feedback. Based on the reviews, we have made several important improvements to the manuscript which are outlined individually below. We hope you will find that the manuscript addresses the concerns of the reviewers satisfactorily.

Once again, our research team is grateful for the ongoing interest of the PLOS ONE in our work. 

Best wishes,

Nathan Coppersmith, MD, MHS

Yale School of Medicine

Yale University 

New Haven, CT 

Editor:

This is an interesting piece of work demonstrating how simple attitudes, such as reducing the number of prescribed opioid tablets in the postoperative period, can result in lower long-term usage risks. 

• Thank you for this feedback. 

I find the study to be well-designed and presented; however, there is a need to enhance the discussion by comparing the obtained results with factors determining postoperative chronic opioid usage, as well as local characteristics of opioid abuse found in official local statistics and any available literature. I believe the discussion is incomplete and does not fully reflect the interesting findings demonstrated.

• Thank you for this feedback. A discussion has been added regarding the risks of developing chronic postoperative opioid usage. Furthermore, the discussion has been enhanced to look at the local and regional trends regarding opioid prescribing habits. 

Reviewer #1: 

Dear author, although this work is very interesting and important to the scientific community, specially dealing with opioid addiction, there are some considerations that I would like to highlight: Opioid consumption must be related to the type of surgery. I have not seen, in the table, this variable. Probably, would be a heterogeneous group, and tissue trauma can vary as well as the intensity of pain.

• Thank you for this feedback. Due to the heterogeneity of surgeries performed across the 191,379 surgical procedures performed during this time period we did not break the data down by type of procedure performed. With such a large variety of surgeries performed, we were more interested in the overall trend of opioid prescription across the healthcare system because change in volume or prescription habits to one or a few particular types of surgeries would not influence the overall trend of such a large volume of surgeries. 

In addition, there was no control of the use of other analgesics that could increase or reduce opioid consumption. Besides that, gender, race, social, educational and economic level, anxiety and other psychological and psychiatric comorbidities and other variables were not controlled.

• Thank you for this feedback. The trend over time of use of other analgesics that could influence post-op opioid consumption is a very interesting and valuable idea. This data was not included in our data pull. There were not large changes in the demographics of gender and race in the population over the study period as demonstrated in the supplementary table 1. The influence of other psychological and psychiatric comorbidities in the population is an interesting point and a data point that was not collected in our data pull. 

There was no mention of risk stratification and the impact on opioid consumption

• Thank you for this feedback. We have incorporated a discussion of patients who are higher risk for developing persistent postoperative opioid use into the discussion as well as risk of worse postoperative outcomes based on prior opioid use. 

I think that the use of opioid have to be a rational, based on specific circumstances, and you have to discuss and suggest how to deal with a well done postoperative analgesia . Reducing the number of pills, or the the need for more over time can not be seen as an isolated event nor as a solution to the opioid crisis

• Thank you for this feedback. We concur that reducing the number of pills is not in isolation a solution to the opioid crisis. The discussion has been enriched to further elaborate on multi-modal pain control strategies, intra-operative pain control strategies, and targeting pill counts to specific procedures that can all be combined in order to potentially reduce the use of post-operative opioids. 

Reviewer #2: 

The manuscript describes a practical synthesis of effect simply reducing the number of opioid analgesic pills in the treatment of postoperative pain in the period 2017 until 2021.

Any rational for reducing 30 pills to 12 pills? why not reduces 30-50% to opioids pill? Can you explain us more?

• Thank you for this feedback. 12 pills was chosen as a quantity to provide one tablet every six hours for three days. This decision was made in 2017 by the healthcare system opioid stewardship committee. This has been added to the manuscript. 

Page 5: Did not mention the calculation for converting each analgesic to morphine (MME), as depending on the reference, the values vary for the calculations.

• Thank you for this feedback. The calculation for converting each analgesic to morphine (MME) was based on the United States Centers for Disease Control and Prevention morphine milligram equivalent doses for commonly prescribed opioids for pain management (https://www.cdc.gov/mmwr/volumes/71/rr/rr7103a1.htm#T1_down). This has been added to the manuscript. 

Page 6: Some studies/papers report a cut off of 80-90 MME, why did they choose 90 MME? There is no consensus among the opioids guidelines as to which dose of morphine would be a cut off, most cite 90 mme.

• Thank you for this feedback. The cutoff of 90 MME was used as it is the equivalent of the default quantity of 12 oxycodone 5 mg immediate-release tablets using the MME conversion of factor of 1.5 for oxycodone based on the CDC MME table. 

Page 6: It was not mentioned which surgeries were included in the research, we know that there are surgeries that have greater painful potential than others. As a suggestion too, include the percentages of outpatient surgeries or separate them by surgical specialty

• Thank you for this feedback. As this included all surgeries performed in the healthcare system, we evaluated the data at a macro level, and we did not evaluate particular surgeries as we felt that the quantity of surgeries is large enough that individual types of surgeries would not influence the overall trend significantly. An interesting follow up study would be to interrogate the trends in prescribing habits of the most frequently performed surgeries, which surgeries had the greatest decrease in post-op opioids prescribed, if any actually had an increase, and which may not have had a decrease.

---

## [Editor Report · Decision Letter 1]

7 May 2024

The persistent benefits of decreasing default pill counts for postoperative narcotic prescriptions

PONE-D-23-43929R1

Dear Dr. Coppersmith,

We’re pleased to inform you that your manuscript has been judged scientifically suitable for publication and will be formally accepted for publication once it meets all outstanding technical requirements.

Kind regards,

Guilherme Antonio Moreira de Barros, M.D., M.Sc., Ph.D

Academic Editor

PLOS ONE

Additional Editor Comments (optional):

Dear Dr. Coppersmith,

Congratulation for the great improvement and for the answers to the questions formulated by the reviewers.
---

## [Editor Report · Acceptance letter]

24 May 2024

PONE-D-23-43929R1 

PLOS ONE

Dear Dr. Coppersmith, 

I'm pleased to inform you that your manuscript has been deemed suitable for publication in PLOS ONE. Congratulations! Your manuscript is now being handed over to our production team.

Kind regards, 

on behalf of

Dr. Guilherme Antonio Moreira de Barros 

Academic Editor

PLOS ONE